# The Association between Fasting Glucose and Sugar Sweetened Beverages Intake Is Greater in Latin Americans with a High Polygenic Risk Score for Type 2 Diabetes Mellitus

**DOI:** 10.3390/nu14010069

**Published:** 2021-12-24

**Authors:** María Lourdes López-Portillo, Andrea Huidobro, Eduardo Tobar-Calfucoy, Cristian Yáñez, Rocío Retamales-Ortega, Macarena Garrido-Tapia, Johanna Acevedo, Fabio Paredes, Vicente Cid-Ossandon, Catterina Ferreccio, Ricardo A. Verdugo

**Affiliations:** 1Programa de Genética Humana, ICBM, Facultad de Medicina, Universidad de Chile, Santiago 8380000, Chile; mllopezportillo@gmail.com (M.L.L.-P.); eduardotobar@ug.uchile.cl (E.T.-C.); milo.yanez20@gmail.com (C.Y.); rmretama@gmail.com (R.R.-O.); 2Departamento de Ciencias Preclínicas, Facultad de Medicina, Universidad Católica del Maule, Talca 3460000, Chile; leahuidobro@gmail.com; 3Advanced Center for Chronic Diseases, ACCDiS, Santiago 8380000, Chile; cferrec@med.puc.cl; 4Facultad de Medicina, Pontificia Universidad Católica de Chile, Santiago 8328888, Chile; magarridot@uc.cl (M.G.-T.); johanna.acevedo.romo@gmail.com (J.A.); fabio.paredes.p@gmail.com (F.P.); vicente.cid@usach.cl (V.C.-O.); 5Departamento de Oncología Básico-Clínica, Facultad de Medicina, Universidad de Chile, Santiago 8380000, Chile

**Keywords:** sugar-sweetened beverages, fasting glucose, genetic risk score, type 2 diabetes mellitus, nutritional epidemiology, genotype by environment interaction, Latin American ancestry

## Abstract

Chile is one of the largest consumers of sugar-sweetened beverages (SSB) world-wide. However, it is unknown whether the effects from this highly industrialized food will mimic those reported in industrialized countries or whether they will be modified by local lifestyle or population genetics. Our goal is to evaluate the interaction effect between SSB intake and T2D susceptibility on fasting glucose. We calculated a weighted genetic risk score (GRSw) based on 16 T2D risk SNPs in 2828 non-diabetic participants of the MAUCO cohort. SSB intake was categorized in four levels using a food frequency questionnaire. Log-fasting glucose was regressed on SSB and GRSw tertiles while accounting for socio-demography, lifestyle, obesity, and Amerindian ancestry. Fasting glucose increased systematically per unit of GRSw (β = 0.02 ± 0.006, *p* = 0.00002) and by SSB intake (β[cat4] = 0.04 ± 0.01, *p* = 0.0001), showing a significant interaction, where the strongest effect was observed in the highest GRSw-tertile and in the highest SSB consumption category (β = 0.05 ± 0.02, *p* = 0.02). SNP-wise, SSB interacted with additive effects of rs7903146 (TCF7L2) (β = 0.05 ± 0.01, *p =* 0.002) and with the G/G genotype of rs10830963 (MTNRB1B) (β = 0.19 ± 0.05, *p* = 0.001). Conclusions: The association between SSB intake and fasting glucose in the Chilean population without diabetes is modified by T2D genetic susceptibility.

## 1. Introduction

Sugar-sweetened beverages (SSB) have high quantity of rapidly absorbed sugars and poor nutritional value [1]. Chile is one of the highest per capita consumers of sugar-sweetened beverages (SSB) at international level [2,3]. However, the biological impacts from such frequent consumption in this population and how these may be modified by its particular genetic composition is unknown. Chileans are largely of admixed ancestry from European (55%), Amerindian (42%), and African (3%) origin [4]. Amerindian ancestry is related to the Aymara and Mapuche ethnic groups and proportions depends on latitude [5]. Further, the population is structured in a way that has preserved a socio-economic cline associated with ancestry [6].

SSB consumption has been related to increased blood insulin concentration, impaired fasting glucose (IFG) [7,8], glucose intolerance (GI), insulin resistance (IR) [9,10,11] and high risk of type 2 diabetes (T2D) [12,13,14]. IFG is considered as a physical state of evolution towards T2D [15], so its control is essential to prevent the T2D onset and cardiovascular diseases [16]. There is also evidence that SSB promote weight gain and increased adiposity due to their high energy content and because they promote higher caloric intake [17,18,19].

Genome-wide studies (GWAS) have found more than 140 T2D variants in independent loci but with a modest association that even in combination explain a low proportion (~20%) of disease liability [20]. Presence pervasive genotype-by-environment interactions (GxE) and rare alleles with large effects, which would be missed from GWAS focusing in common variants, have been presented as possible explanations for the missing heritability [21,22]. However, large whole-genome and whole-exome efforts have found only a handful of rare variants that explain even a smaller fraction of phenotypic variation [23,24]. The relevance of GxE in predisposition to T2D could be used to detect groups at higher risk and indicate that interventions are best conceived in a population-specific or even individual-specific manner [25].

There is interest in identifying genetic loci implicated in fasting glucose and lifestyle-gene interactions that modify the incidence of T2D because they may suggest relevant biological pathways [26,27]. Interaction between fruit intake and polygenic genetic risk for type 2 diabetes have shown an effect on both diabetes risk and on fasting glucose [28]. Similar studies have explored interaction with SSB, without conclusive results [29]. A study on subjects of European origin evaluated the interaction of a T2D-GRS and foods related to T2D risk (including sugar-sweetened beverages) but did not have significant results [30]. A meta-analysis of 11 cohorts including 34,748 participants without type 2 diabetes that included loci related to the CHREB pathway of fructose metabolism found a significant association of SSB consumption with insulin concentration and fasting glucose, but no interaction on was found [8].

However, consumption of carbohydrate-rich foods has shown interaction with individual T2D-risk variants, among which Transcription Factor 7 Like-2 (TCF7L2) has been consistently replicated [25]. For instance, a prospective study including 5477 participants of European origin found that a higher intake of whole grain was associated with a 34% lower risk to deteriorate in glucose tolerance to prediabetes or T2D. They show that the protective effect of whole-grain intake was lost in subjects who carried the T allele of rs7903146 in TCF7L2 [31]. Recently, in a Swedish case-control study conducted in a European population investigate whether the association between sweetened beverages and T2D is modified by TCF7L2 but there were no indications of synergistic effects [32].

The present study aims to evaluate the association between sugar-sweetened beverage consumption and fasting glucose, and to establish whether this relationship is affected by Chilean genetic variation. In a sample of subjects participating in the MAUCO cohort without T2D self-reporting, we investigated the presence of interaction effects between sugar-sweetened beverage consumption and T2D genetic susceptibility considering ancestry and obesity. We provide the first genotype-SSB assessment of interaction effect on fasting glucose in a Latin American population.

## 2. Materials and Methods

### 2.1. Study Population and Data Collection

MAUCO is a prospective cohort study initiated in 2014 in the city of Molina, Maule Region, 200 km south of Chile’s capital Santiago in which 8970 eligible participants were enrolled. The aim of this project is to study the natural history of some chronic diseases and Cancer and associated risk factors. In brief, all adult inhabitants of Molina were invited to participate. A baseline health interview was applied at home, where a validated food frequency questionnaire of Mediterranean diet was applied [33].

The survey also assessed demography, socioeconomic status, educational level, lifestyle factors such as frequency of 30 min of physical activity (three or more times per week, once or twice per week, less than 4 times per month), smoking status, and alcohol use. Subsequently, all participants were called to a health station to collect anthropometric, biochemical measurements and collection of biological samples (urine, saliva, and blood). Waist circumference (WC) was measured with a non-distensible fiberglass tape and was classified as normal or high according to the Adult Treatment Panel III (ATPIII) criteria. Weight and height were measured with a SECA 700 scale with height rod and a up to 200 cm. Body mass index (BMI) > 25 and <30 points were classified as overweight and those with 30 or more as obese. The biochemical measurements obtained for this analysis were HDL cholesterol (mg/dL) and triglycerides (mg/dL), which were classified as normal or low/high according to ATPIII criteria. Fasting glucose was measured in venous blood after 8 to 12 h of fasting using the glucose oxidase method in a CT 600i spectrophotometer (Weiner Lab, Rosario, Argentina). The detailed methodology is described elsewhere [33].

We conducted a cross-sectional study with a sample of 3321 subjects aged 36–77 years participating in the MAUCO cohort without self-reported T2D. subjects with a genotyping rate of less than 80% of selected SNPs and without SSB consumption register were eliminated. The final sample was 2828 subjects with complete genomic and environmental information. A workflow representation of sample sizes and data filtering is shown in Figure 1. All participants who met the criteria information for this investigation signed the informed consent form. This research was approved by the Ethics Committee of the Pontificia Universidad Católica de Chile and the Ministry of Health of the Maule Regional Health Services.

### 2.2. Evaluation of Sugar-Sweetened Beverage Consumption

Dietary information was obtained with a semi-quantitative food frequency questionnaire of Mediterranean diet applied by previously trained personnel. The average consumption per day during the last year was specified according to 4 levels as follows: category 1:no consumption (0 = 0), category 2: >0 and <1 servings/day, category 3: 1≤ and <2 servings/day, and category 4: ≥2 servings/day. One serving is equivalent to 350 mL. Sugar-sweetened beverages were defined by containing caloric sweeteners such as sucrose or high fructose corn syrup, providing the following examples: regular soda, industrialized fruit juices with natural or artificial ingredients, energy drinks, and sports drinks. “Light”, “zero”, “low”, or “reduced calorie” beverages as well as natural juices were excluded. High-risk alcohol consumption was defined by weekly servings of ≥30 for men ≥20 for women.

### 2.3. Selection of T2D Risk SNPs and Ancestry

Single nucleotide polymorphisms (SNPs) for T2D risk were obtained from a Genome Wide Association Studies (GWAS) for T2D and from meta-analyses. SNPs were selected by statistical significance of ≤3 × 10^−8.^ From these, the set of not redundant (not in LD) were selected, while prioritizing those with the largest OR (odds ratio) and those that were assessed in populations of Latin American ancestry. A total of 21 SNPs were selected, of which 17 resulted in successful PCR assays (Appendix A).

### 2.4. Genotyping

The 17 T2D SNPs were genotyped separately from 150 ancestry informative markers (AIMs) in two multiplexed reactions using a modified version of GT-Seq protocol [34]. Briefly, genomic DNA was distributed in 96-well plates and amplified using the PCR (polymerase chain reaction) technique (PCR1) with locus-specific primers. A second reaction (PCR2) added Illumina sequencing adapters and index pairs to uniquely identify wells and plates. Next, amplicon concentration across wells was homogenized using SequalPrep™ Normalization Plate Kit (Applied Biosystems, Waltham, MA, USA) before mixing samples into one pool per plate. Finally, an equimolar mixture of pools was created and sequenced in a NextSeq 550 (Illumina Inc., San Diego, CA, USA) with Mid Output v2 kits in 1 × 125 configuration. Genotypes for 17 T2D SNPs and 150 AIMS were successfully obtained but rs188827514 was excluded from the analysis because it had a minor allele frequency less than 5%.

### 2.5. Bioinformatic Analysis

The *bcl* files generated were converted to a *fastq* format with *bcl2fastq* v2.20.0.422 and quality-controlled with the *fastqc* v11.5. Primer homodimers and heterodimers were removed and one *fastq* file per sample was generated according to the sequence of the i7 (plate) and the i5 (well) indices by using scripts available at https://github.com/GTseq/GTseq-Pipeline (accessed on 26 January 2018). On-target reads, were defined as having both the sequence of locus-specific forward primer and a known sequence of 17 base pairs around the selected SNP. The genotype assignment was performed by counting reads matching to each allele of the selected SNPs in each sample.

### 2.6. Ancestry Inference

A set reference dataset with 224 samples was created by joining whole-genome sequencing (WGS) data for 30 individuals of African (YRI) ancestry and 30 of European (CEU) ancestry from the 1000 Genomes Project and Axiom LAT1 microarray data from 110 Aymaras from Puno, Peru (Andrés Moreno-Estrada, personal communication), 54 high Amerindian ancestry from central and southern Chile from the ChileGenomico and PatagoniaDNA projects [5,35]. Global ancestry proportion per individual was calculated with the ADMIXTURE program (version 1.23) counting the reference group allele frequencies and those of the mestizo sample under study. A script was implemented to calculate the ancestry proportion in thousands of individuals by creating bins of 50 MAUCO participants in order to minimize bias due to the large sample size of this dataset.

### 2.7. Polygenic Risk Score Calculation

All T2D SNPs that met the quality criteria were included to construct a weighted Genetic Risk Score (GRSw) as follows: first, a count of T2D-associated risk alleles of each polymorphism was performed (0 = protective homozygous, 1 = heterozygous, 2 = risk homozygous). The GRSw calculation was performed adding the Beta effect (β) obtained from the natural logarithm of the odds ratio (OR) reported in genome-wide studies (GWAS) performed in Latinos or multi-ethnic studies that considered this ancestry, multiplied by the number of risk alleles carried by each subject as follows: GRSwi = $\sum_{j=1}^{k}\beta_{j}x_{i}$, where *i* is a subject index, *k* is the number independent SNPs genotyped in each subject, $\beta_{i}$ is the estimated effect for the risk allele of each SNP *j* and $x_{i}$ is the number of risk alleles in subject *i*. $\beta_{i}$ was obtained from publications. The odds ratio (OR) was reported, $\beta_{i}$ wasestimated as $\ln\left(OR\right)$. If the OR for the protective alleles was reported, $\ln\left(1/OR\right)$ was used.

### 2.8. Statistical Analysis

The output variable (fasting glucose) was log transformed in base e in order to approximate a normal distribution. Univariate association with categorical variables was tested by the Chi-square tests. Association with continuous variables was assessed with the Student’s *t*-test, ANOVA or Kruskal–Wallis for non-parametric variables.

Association between fasting glucose-log and SSB, GRSw and individual SNPs was evaluated using multiple linear regression models, while adjusting for age, sex, waist circumference, physical activity, educational and socioeconomic level, smoking status, high consumption of fruits, vegetables, sugar, processed meats, and alcoholic beverages (as defined in Table 1), Amerindian ancestry tertile, and GRSw. GRSw was included both as a continuous variable and as a categorial factor by classifying individuals into tertiles of GRSw. All two-way interaction terms between sex, SSB, Ancestry, and GRSw where considered. When testing interaction between SSB and individual SNPs, no GRSw term was included and all two-way interaction terms between sex, SSB, Ancestry, and SNP where considered. One SNP at a time was included. *p* values were calculated from t-tests for individual each regression coefficient (H_0_:β = 0) and from F-tests for each factor by comparing the full model with a reduced model that dropped a single term using the drop1 function in R. Significance was set at 0.05 and Bonferroni correction was used when testing association and interactions for individual SNPs only. The proportion of variance explained by each SNP was calculated by *R*^2^ = SS_SNP_/SS_Tot_, were SS_SNP_ is the sum of squares due to the SNP’s marginal and interaction effects and SS_SNP_ is the total sum of squares, i.e., $\sum_{i=1}^{n}{\left(y_{i}-\bar{y}\right)}^{2}.$

Post-hoc power estimation to detect (replicate) genotype-to-phenotype associations were performed using the online web tool (https://clincalc.com/stats/power.aspx, accessed on 14 December 2021) based on the marginal effects of association between SNPs (categorical) and fasting glucose (log) (continuous). All statistical analyses were performed with RStudio version 1.31093.

## 3. Results

### 3.1. General Characteristic of the Study Population

The studied sample consisted of 3866 subjects aged 36 to 77 years, 32.2% (*n* = 1247) are male and 67.8% (*n* = 2619) are female with an average age of 54(9.1) years of which 42.7% (*n* = 1652) with overweight (SP) and 40.9% (*n* = 1580) with obesity (OB) so the combined prevalence of SP and OB is 83.6% (*n* = 3247). The prevalence of central obesity measured by waist circumference was 71.2% (*n* = 2754) and 14.1% (*n* = 545) reported being diabetic or taking glucose-lowering medication.

The final sample size including complete genomic, anthropometric, socio-demographic and lifestyle information was 2828 subjects. General characteristics and their association with fasting glucose in the whole sample and stratified by sex are shown in Table 1 and Appendix A.

### 3.2. General Characteristics by SSB Consumption

Results show significant differences in general characteristics of participants by intake level. High SSB consumption was more frequent in men (*p* < 0.0001) and in younger individuals (*p* < 0.03), active smokers (*p* < 0.001), and risky alcohol drinkers (*p* < 0.0001). In addition, consumers of sugar-sweetened beverages reported lower consumption of fruits and vegetables (*p* < 0.0001) and higher intake of sugar and processed meats in their diet (*p* < 0.0001). Progressive increase on fasting blood glucose-log and WC are observed for each increase in SSB category of consumption (*p* < 0.0001) (Appendix A).

SSB consumption was associated with higher log-fasting glucose when comparing the highest (≥2 servings/day) versus the lowest category (0 servings/day) of intake in the whole sample (β = 0.04 ± 0.01, *p* < 0.0001) and in women (β = 0.05 ± 0.01, *p* < 0.001). When BMI was excluded from models, the association remained significant in the whole sample (Appendix A). No significant association was observed in males. There was also no evidence of a sex-by-SSB interaction (*p* > 0.51) (Appendix A).

### 3.3. Fasting Glucose and SNPs: Association and Interaction Effect

Among the 17 selected SNPs, rs188827514 (CCND2) was excluded because it had a MAF (Minor Allele Frequency) lower than 5%. The association between each SNPs that had significant association with T2D in GWAS studies and fasting glucose-log. SNPs were located in 12 different chromosomes, and all are distant enough so that they are nonredundant (r^2^ < 0.80).

Our results showed a positive association between log-fasting glucose and variants rs516946 in *Ankyrin 1* (ANK1), rs4402960 in Insuline-Like Growth Factor 2 mRNA Binding Protein (IGF2BP2), and rs7903146 in Transcription Factor 7 Like 2 (TCF7L2) genes, while adjusting for age, sex, BMI and Amerindian Ancestry, after Bonferroni correction (*p* < 0.0004, 0.0007 and 0.0009, respectively) (Appendix A). The association remain significative when BMI is excluded from the model, indicating they are not mediated by adiposity.

An interaction effect was found between rs7903146 (TCFL2) (as continuous) and the highest SSB category (β= 0.05 ± 0.01, *p* < 0.002) (Appendix A). When regressed by number of risk alleles, significance was lost after Bonferroni correction (*p* < 0.005) (Table 2 and Appendix A). When genotypes were coded as a categorical variable, rs10830963 Melatonin Receptor 1B (MTNR1B) showed significant positive interaction between the G/G genotype of rs10830963 (MTNR1B) variant and the highest category of SSB consumption (cat4) on log-fasting glucose in the overall sample after Bonferroni correction (β = 0.19 ± 0.05, *p* < 0.001). Post-hoc power estimations showed that our sample size provided ≥80% power to detect effects of OR ≥ 1.01 for genotypes with high frequency genotyping rate, but it was limited for other SNPs (Appendix A). Results were unaffected when BMI was excluded from the models (data not shown). The interaction results for these two SNPs are found in Table 2 and for all SNPs in Appendix A.

### 3.4. Fasting Glucose and GRSw: Association and Interaction Effect

We constructed an effect-weighted risk score from the 16 selected SNPs of T2D risk. β estimates for association (used in the score) came from studies including Latin American participants whenever possible. The GRSw had an average value of 4.16 (SD = 0.48, min 2.02, max 5.85) and a gaussian distribution (not shown). Linear regression accounting for demographics, lifestyles and ancestry showed that fasting glucose-log increased 0.02 per unit of GRSw (β = 0.02 ± 0.006, *p* < 0.00002) in the whole sample, i.e., increased 1.02 mg/dL for each GRSw increment (Table 3). When GRSw was discretized in tertiles, participants in the highest tertile incremented log-glycemia also in 0.2 versus the lowest tertile (β = 0.02 ± 0.007, *p* < 0.0003) (Table 3). The effect was marginally larger in females (β = 0.03 ± 0.008, *p* < 0.0024) than in males (β = 0.02 ± 0.01, *p* < 0.08) but no significant interaction was observed between sex and GRSw (Appendix A). Results remained significant when BMI removed from the models (data not shown).

We observed a tendency of increasing effects from SSB consumption as the tertile of GRSw increased (Figure 2). There was a significant and positive interaction between continuous SSB and continuous GRSw (β = 0.02 ± 0.006, *p* < 0.004) (Table 4). That is, the increment in log-glycemia by each level of consumption is 2% higher with every unit of increment in GRSw (e^0.02^ = 1.02). When SSB was considered as categorical variable, the trend by GRSw was higher in the highest category of SSB consumption in the whole sample (β = 0.06 ± 0.02, *p* < 0.01) and in men (β = 0.08 ± 0.04, *p* < 0.03) (Table 4). Although the interaction was significant only in the global sample and in men, we did not detect a significant interaction of sex with either SSB or GRSw (Appendix A). When both SSB and GRSw where rendered categorical, the interaction effect was only detected in the highest levels of SSB consumption (cat4) and GRSw (tertil3) in the whole sample (β = 0.05 ± 0.02, *p* = 0.02). The interaction was also observed in men (β = 0.07 ± 0.05, *p* = 0.009) (Table 4). The effect modification remained when including or excluding BMI as a covariate (not shown).

### 3.5. Sensitivity Analysis

After excluding subjects older than 65 years old, who were at higher risk because age is a risk factor, the interaction effects were attenuated. However, the interaction between sugary drinks consumption and GRSw (as continuous) was preserved (*p* < 0.04). Similarly, the *P trend* of the interaction between SSB consumption and rs7903146 (TCF7L2) was significant in double risk allele carriers (*p* < 0.003), but significance is lost after Bonferroni correction (*p* < 0.03). In a second analysis, including two levels of consumption were considered (less than 1 serving/day = 0, and 1 or more servings/day = 1) in order to balance sample sizes. Interaction remains significant, and was accentuated in the subjects belonging to the highest category of consumption (Appendix A).

## 4. Discussion

Here we preset results from a cross-sectional study that included 2828 participants from the MAUCO cohort without self-reported T2D diagnosis. The positive association of sugar-sweetened beverages and fasting glucose has been inconsistent in the international literature. However, majority of studies and a recent meta-analysis tended to support the unfavorable effect of these beverages on glycemic traits, including fasting glucose [7,8,36].

In agreement with previous studies our results indicate a positive association between SSB consumption and fasting glucose when adjusting for all possible confounders. Furthermore, significance was preserved when adjusting for GRSw and proportion of Amerindian ancestry. The general characteristics clearly showed that consumption of fruits and vegetables decrease while that of processed meats increase when SSB intake is higher. The same trend of increase was observed on fasting glucose levels, obesity, waist circumference and triglycerides. HDL-c cholesterol decrease when SSB consumption increases.

We used a weighted polygenic risk score (GRSw) based on 16 T2D SNPs previously observed in Latinos. Most of them have been related to HOMA-β which is an indirect measure of the first phase of insulin secretion [37,38,39]. Results showed linear association between GRSw and fasting glucose while accounting for clinical, socio-demographic, and lifestyle factors, as well as Amerindian ancestry. The highest glucose levels correspond to subjects with the highest DM2 genetic risk. That is, the score increases as blood glucose rises, so is therefore considered to be a useful tool for measuring genetic susceptibility.

SSB-FG association was intensified in subjects with the highest consumption and the highest GRSw. In other words, the association between SSB consumption and fasting glucose was amplified in subjects with the highest genetic susceptibility that consume two or more servings of SSB per day. All our results were not change when BMI was excluded from models, suggesting that this genetic effect is not mediated by obesity.

When analyzing each selected SNP, we found association between rs7903146 (TCF7L2), rs4402960 (IGF2BP2), rs516946 (ANK1) and fasting glucose. TCF7L2 is a transcription factor located in the non-coding region of Chromosome 10 and the association with fasting glucose was previously observed in Chileans by Peterman and coworkers [40]. This variant have the greatest effect on T2D risk and is recognized as the most important regulator of pro-insulin expression and processing [41]. It has been observed that TCF7L2 influences regulation of glucose metabolism through the Wnt signaling pathway [42] and that carriers of the double risk allele (T/T) were more likely to have progression from impaired glucose tolerance to diabetes than were CC homozygotes and decreased insuline secretion [43]. IGF2BP2 (insulin like growth factor 2 MRNA binding protein 2) gene encodes a protein that binds the 5′ UTR of insulin-like growth factor 2 (IGF2) mRNA and regulates insulin translation. This variant increase T2D risk through an insulin secretory mechanism, including the lower first-phase insulin response [44]. ANK1 is a protein coding gene called *ankyrin-1* expressed mainly in red blood cells and has been associated with HbA1c (glycated hemoglobin) [45]. ANK1 has been found in skeletal muscle but the mechanism by which sAnk1 in skeletal muscle might be involved in T2D remains uncertain [46].

When considering diet-dependent genetic effects, we found positive interaction between SSB consumption and T2D-SNPs on fasting glucose (FG). In two of the sixteen selected SNPs, rs7903146 (TCF7L2) and rs10830963 (MTNR1B), a positive interaction with SSB was observed. The largest effect was observed among homozygous risk carriers belonging to the highest category of SSB consumption. This research reports the first SSB-genotype interaction analysis in Latin American subjects. Therefore, this study represents an approximation to the knowledge of the genetic risk on glycemic balance and the genes involved in Latinos.

The first scientific evidence on diet-gene interaction has been provided by studies that have employed BMI risk scores to measure the genetic effect on adiposity level [47,48,49]. There are few studies that have evaluated the food-gene interaction when the outcome is T2D or its quantitative traits. In a prospective nested case-control study including 1196 diabetic and 1337 non-diabetic men a significant interaction (*p* < 0.02) was observed between T2D-GRS and Western dietary pattern on T2D risk. The effect were more evident among men with high genetic score and the highest quartile of Western dietary pattern [50]. Ericson and coworkers examined the interaction between a GRS constructed from T2D-48 SNPs, and a diet risk score of four foods associated with T2D (processed meat, sugar-sweetened beverages, whole grain and coffee) in 25,069 individuals of European origin, but no significant interaction between the GRS or food components was observed [30].

Previous research showed that variants in MTNR1B were consistently associated with fasting glucose and the strongest signal was observed at rs10830963, where each G allele was associated with an increase in fasting glucose and reduced beta-cell function [51]. A meta-analysis evaluating whether diet modifies the association between circadian-related variants and cardio-metabolic risk factors subjects of European origin showed the protective association of a higher carbohydrate intake in rs1387153 (MTNR1B) was weaker for each additional T allele. That is, the T allele attenuates the inverse association between carbohydrate intake and FG [52]. Finally, in a randomized cross-over trial evaluating glucose-tolerance in late-dinner and early-dinner of homozygous carriers and non-carriers of the MTNR1B risk allele suggest that moving the dinner to an earlier time may result in better glucose-tolerance specially in MTNR1B carriers [53].

Other researchers have also been interested in testing SNP-by-SSB interaction. A recent meta-analysis conducted by the CHARGE Consortium in 11 cohorts evaluated the effect of SSB intake and 18 genetic variants in 11 genes related to fructose metabolism (ChREB pathway) on glycemic traits in individuals of European descent. They only found a suggestive interaction between KLB (rs1542423) and SSB consumption on fasting insulin, but not other significant interactions on fasting glucose were found [8]. In parallel, a Swedish case–control study including 1253 cases and 1545 controls evaluated the association and interaction effects of rs7903146 and SSB consumption on T2D risk [32]. Although they found a positive association across genotypes, no evidence of interaction effects was found.

Strengths of our study lie in the large number of socio-demographic and lifestyle variables collected by highly trained personnel producing dietary and environmental information with few missing data. The high consumption of SSB (exposure factor) in the Chilean population facilitated assessing its association with the outcome variable. Limitations lie in the low number of genetic loci that were evaluated and in the potential measurement errors of SSB consumption since it was indirectly estimated though a food frequency questionnaire and the relatively limited sample size. Post-hoc power equal showed that our sample size was sufficient to detect associations with an OR ≥ 1.01, depending on allele frequency and genotyping success rate. However, the genetic risk score was constructed with a small proportion of polymorphisms explaining only a small proportion of the genetic risk which may represent a limitation in the estimation of genetic susceptibility.

The fact that we did not identify an association between Ancestry and fasting glucose must not be overinterpreted. This may result from the little ancestry variation present in the MAUCO cohort, reducing its power to detect an association. In addition, Amerindian ancestry was positively and significantly associated with SSB consumption, probably mediated by differences in socio-economic level, indicating that any test for association with ancestry must also consider these other variables to avoid confounding effects. It is important to consider a possible bias and overestimation of the effect size from SSB since it may be associated with other food components of habitual diet as well as detrimental lifestyle factors not included in the survey or models. Furthermore, because the exposure and the outcome were measured at a specific time it is not possible establish a causal relationship.

Despite the described limitations, our study provides the first evidence of genotype-by-diet interaction in a Latin American population that modifies the risk for T2D before clinical sings of the disease. Further studies will be necessary to confirm our findings and comprehensive genomic profiles will be needed to identify genetic variation that may affect T2D risk factors in this population. These types of studies will contribute to devising new hypothesis for potential mechanisms of genetic effects on increasing susceptibility to diabetes and may propose targeted public-health strategies to control the diabetes pandemic in Latin America.

## 5. Conclusions

This cross-sectional study provides evidence of interaction effects between sugar-sweetened beverages consumption and genetic susceptibility to T2D on fasting glucose in a sample of Chileans who did not report to be diabetic. The adverse effects of SSBs on fasting glucose is observed in those consuming at least two servings per day and is magnified in individuals at the highest genetic susceptibility level in a non-adiposity-mediated manner. The biological pathways triggered by SSB consumption may involve the TCF7L2 and MTNR1B genes.

## Figures and Tables

**Figure 1 nutrients-14-00069-f001:**
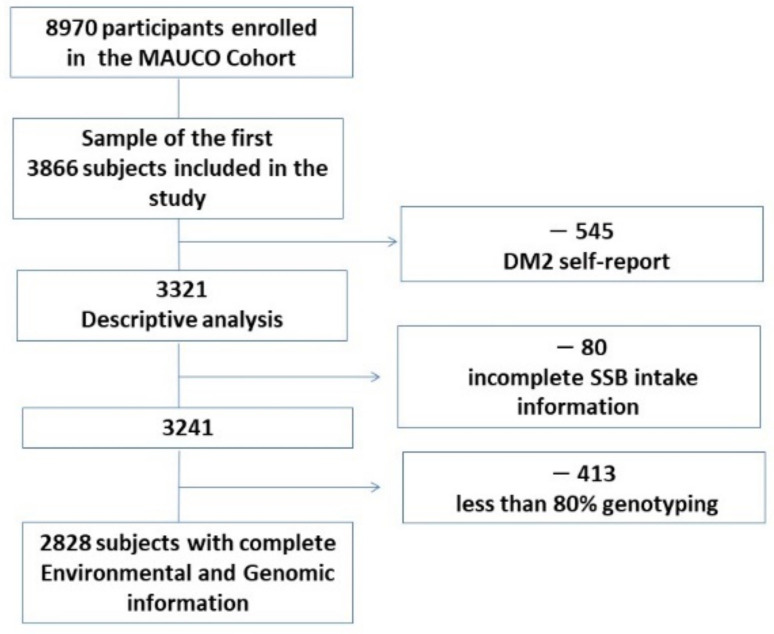
Number of participants and data filtering steps.

**Figure 2 nutrients-14-00069-f002:**
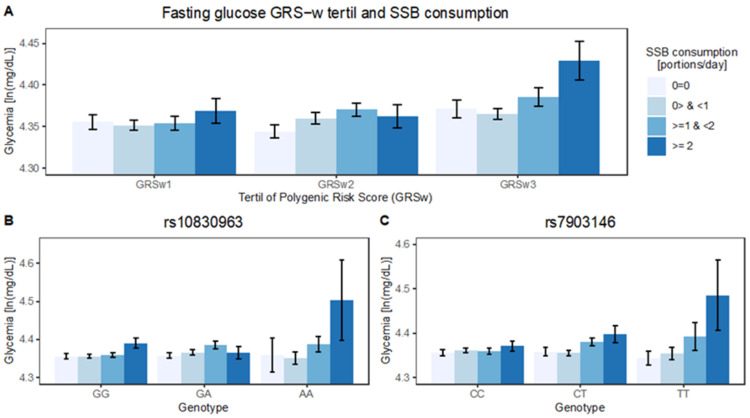
Fasting glucose association with SSB is dependent on genetic risk for T2D. Average log-fasting glycemia is plotted SSB category against tertiles of GRSw (**A**) and genotypes of rs10830963 (MTNR1B) (**B**) and rs7903146 (TCF7L2) (**C**), *n =* 2828.

**Table 1 nutrients-14-00069-t001:** General characteristics of participants.

	Overall (*n* = 2828)	*p*	Men (*n* = 971)	*p*	Women (*n* = 1857)	*p*
Sex (%)	-		34.3		65.7	
Age (years)	53.4 (9.5)	**0.0001**	54.0 (9.6)	**0.003**	53.1 (9.4)	**0.0001**
Education (% low/medium/high)	49.0/39.2/11.8	0.14	47.4/40.8/11.8	0.34	49.8/38.4/11.8	0.33
Socioeconomic (% low/med/high)	24.3/58.9/16.8	0.13	16.2/60.8/22.1	0.75	28.0/57.9/14.1	0.81
Physical Activity (% active/inactive)	19.96/80.04	0.34	20.0/80.0	0.90	19.97/80.03	0.24
Smoking (% never/current/former)	42.8/33.1/24.1	**0.0001**	32.9/35.9/31.2	0.48	47.9/31.7/20.4	**0.003**
Amerindian ancestry	0.35 (0.08)	0.33	0.34 (0.07)	0.19	0.35 (0.08)	0.46
High risk alcohol consumption (%)	15.3	0.35	27.6	0.57	9.0	0.37
Fruit intake ≥ 1 portions/day (%)	48.3	0.45	38.4	0.63	53.3	**0.03**
Vegetable intake ≥ 1 portions/day (%)	68.7	0.33	60.0	0.07	73.1	0.25
Processed meat ≥ 4 portions/week (%)	10.1	0.72	12.3	0.35	8.6	0.48
Sugar ≥ 4 teaspoons/day (%)	38.8	0.94	47.3	**0.04**	34.4	**0.0001**
SSB category (%) (*n*)						
0 servings/day	24.3 (686)	**0.0001**	13.0 (126)	0.16	30.2 (560)	0.02
>0 and <1 servings/day	37.6 (1064)		38.0 (369)		37.4 (695)	
≤1 and <2 servings/day	25.1 (710)		27.8 (270)		23.7 (440)	
≥2 servings/day	13.0 (368)		21.2 (206)		8.7 (162)	
Glycemia ≥ 100 mg/dL (%) (*n*)	21.9 (618)	-	30.0 (291)	-	17.6 (327)	-
BMI (kg/m^2^)	29.3 (4.8)	**0.0001**	28.9 (4.1)	**0.0001**	29.6 (5.0)	**0.0001**
BMI (normal/overweight/obesity) (%)	16.7/44.2/39.1	**0.0001**	16.6/47.0/36.4	**0.001**	16.7/42.9/40.4	**0.0001**
WC (cm)	98.1 (10.8)	**0.0001**	101.0 (9.4)	**0.0001**	96.5 (11.0)	**0.0001**
High WC (%)	68.9 (1949)	**0.0001**	45.8 (443)	**0.0001**	81.0 (1506)	**0.0001**
Triglycerides (mg/dL)	162.8 (122.4)	**0.0001**	181.6 (153.0)	**0.0001**	153.0 (101.5)	**0.0001**
Hypertriglyceridemia (%)	44.2 (1251)	**0.0001**	50.5 (490)	**0.0001**	41.0 (761)	**0.0001**
HDL-c (mg/dL)	45.8 (11.1)	**0.0001**	42.3 (10.4)	0.19	47.7 (11.0)	0.06
Low HDL-c (%)	44.5 (1258)	0.22	56.2 (544)	0.22	38.4 (714)	**0.01**

Data are expressed as mean and standard deviation (SD) for continuous variables and as percentage (%) and sample size (*n*) for categorical variables. *p*-value for association of each variable with log-fasting glucose are shown. *p*-values in bold mean significance (<0.05).

**Table 2 nutrients-14-00069-t002:** Interaction between rs7903146 and rs10830963 (as continuous and categorical variable) and levels of SSB categories intake on log-fasting glucose.

rsID	Gen	Genotype	SSB Category 2		SSB Category 3		SSB Category 4		P_i_
			Β (SE)	P_t_	Β (SE)	P_t_	Β (SE)	P_t_	
rs7903146	TCF7L2	0,1,2	0.0004 (0.01)	0.97	0.020 (0.01)	0.11	0.05 (0.01)	**0.001**	**0.002**
		C/C	4.37 (0.053)	0.32	4.20 (0.06)	0.44	4.33 (0.072)	0.80	0.005
		C/T	−0.006 (0.02)	0.10	0.03 (0.02)	0.10	0.03 (0.02)	0.09	
		T/T	0.009 (0.03)	0.77	0.02 (0.03)	0.53	0.14 (0.04)	**0.0006**	
rs10830963	MTNR1B	0,1,2	0.005 (0.01)	0.67	0.01 (0.01)	0.43	0.013 (0.01)	0.47	0.83
		C/C	4.21 (0.053)	0.26	4.30 (0.058)	0.61	4.28 (0.072)	0.88	**0.001**
		C/G	0.01 (0.02)	0.54	0.01 (0.02)	0.41	−0.03 (0.02)	0.16	
		G/G	−0.004 (0.04)	0.91	0.01 (0.05)	0.75	0.19 (0.05)	**0.0008**	

β: intercept of the reference (first) genotype or regression coefficient for number of risk alleles and for non-reference genotypes when treated as a categorical variable, SE: standard error, P_t_: *p* value for trend, P_i_: *p* value for interaction. Model adjusted for: age, sex, BMI, waist circumference, physical activity, education, socio-economic level, smoking, consumptions: fruit, vegetables, sugar, processed meats, alcoholic beverages and % Amerindian ancestry, *p* values in bold are significant after Bonferroni correction (*p* < 0.0031), *n* = 2828.

**Table 3 nutrients-14-00069-t003:** Association between GRSw and fasting blood glucose levels (log) as continuous and categorical variable.

	Global		Men		Women	
	β (SE)	*p*	β (SE)	*p*	β (SE)	*p*
GRSw (continuous)						
GRSw	0.02 (0.006)	**0.00002**	0.03 (0.01)	**0.005**	0.02 (0.007)	**0.003**
GRSw (categorical)						
GRSw tertile 2	0.001 (0.006)	0.85	−0.007 (0.01)	0.62	0.004 (0.008)	0.60
GRSw tertile 3	0.02 (0.007)	**0.0003**	0.02 (0.01)	0.08	0.03 (0.008)	**0.002**

Data are expressed in β (standard error), 95% confidence interval (95% CI). Model adjusted for: age, sex, BMI, waist circumference, physical activity, schooling, socioeconomic level, smoking, consumptions: fruit, vegetables, sugar, processed meats, alcoholic beverages and % Amerindian ancestry, *n* = 2828. *p* values in bold are significant (*p* < 0.05).

**Table 4 nutrients-14-00069-t004:** Interaction effects between GRSw and sugar-sweetened beverages intake on fasting blood glucose levels as continuous/categorical variables.

SSB Category	Global			Men			Women		
	β (SE)	P_t_	P_i_	β (SE)	P_t_	P_i_	β (SE)	P_t_	P_i_
SSB and GRSw as continuous									
	0.02 (0.006)	0.004	**0.004**	0.03 (0.01)	0.01	**0.01**	0.008 (0.006)	0.25	0.25
Categorical SSB and continuous GRSw									
2	0.01 (0.01)	0.49	**0.01**	0.003 (0.04)	0.92	**0.03**	0.02 (0.02)	0.26	0.46
3	0.01 (0.02)	0.41		0.02 (0.04)	0.54		0.007 (0.02)	0.71	
4	0.06 (0.02)	**0.001**		0.08 (0.04)	0.04		0.03 (0.02)	0.16	
SSB and GRSw as categorical									
	GRSw Tertile 2								
2	0.03 (0.02)	0.09	**0.02**	0.02 (0.04)	0.62	**0.009**	0.04 (0.02)	0.053	0.57
3	0.03 (0.02)	0.15		0.06 (0.05)	0.22		0.01 (0.02)	0.50	
4	0.0005 (0.02)	0.98		0.03 (0.05)	0.60		0.03 (0.03)	0.26	
	GRSw Tertile 3								
2	0.007 (0.02)	0.70		−0.01 (0.04)	0.52		0.02 (0.02)	0.30	
3	0.02 (0.02)	0.41		0.04 (0.05)	0.38		0.006 (0.02)	0.76	
4	0.05 (0.02)	**0.02**		0.07 (0.05)	0.12		0.04 (0.03)	0.26	

β estimated interaction effect, SE: standard error, P_t_: *p* value for trend, P_i_: *p* value for interaction. Models adjusted for: age, sex, BMI, waist circumference, physical activity, schooling, socio-economic level, smoking, consumptions of fruit, vegetables, sugar, processed meat, and alcoholic beverages, and Amerindian ancestry, *n* = 2828. *p* values in bold are significant (*p* < 0.05).

## Data Availability

Data available on request due to privacy restrictions.

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
