# Peer review of "The Association between Fasting Glucose and Sugar Sweetened Beverages Intake Is Greater in Latin Americans with a High Polygenic Risk Score for Type 2 Diabetes Mellitus"

_nutrients, 2021, doi:10.3390/nu14010069_

Round 1

Reviewer 1 Report

The authors conduct a cross-sectional study and aimed to evaluate the association between SSB intake/T2D-associated risk SNPs and fasting glucose in the Chilean population. The studied sample consisted of 3321 subjects (men, n= 1098 and women, n=2223) without T2D.

Comments:

1.

Table 1, SSB category

Sample sizes are inconsistent for SSB category: 686+1064+710+368+80= 2908<3321 subjects. 

Sample sizes are also inconsistent for SSB category among men (983<1098 subjects) and women (1925<2223 subjects).

Table 1, Glycemia≥100 mg/dl

Sample sizes are also inconsistent for Glycemia≥100 mg/dl:683/0.2318=3129<3321 in overall, 297/0.3=980<1098 in men, 338/0.1765=1916<2223 in women and 980+1916=2896<3129 or 3321 in combined men and women.

The sample sizes should be need to recheck in this study.

2.

SSB consumption has been related to impaired fasting glucose (IFG) [7–9] glucose 47 intolerance (GI), insulin resistance (IR) [10–12] and high risk of type 2 diabetes.

2.3 Selection of T2D risk SNPs

Single Nucleotide Polymorphisms (SNPs) for T2D risk were obtained from a Genome Wide Association Studies of (GWAS) of T2D from previous GWAS and meta-analyses 135 (SNPs were selected by statistical significance of ≤3E-8.  

Similarly, selected T2D-associated risk SNPs are instrumental variable (Mendelian randomization).

Maybe they have been clearly investigated the relationship of fasting glucose.

3.

Authors showed the effect modification (interaction) of SSB consumption and T2D-associated risk SNPs on the fasting glucose.

Additionally, effect size or attributable risk should be reported (proportion of variance explained).

4.

Table 2

TCF7L2 rs7903146 Genotypes shown 2 “T/T”; it should be typing error.

The TCF7L2 rs7903146 “T/T”(maybe C/C) shows the value of 4.43 in SSB Category 2, 4.20 in SSB Category 3………………………….

The “C/T” shows the value of -0.006 in SSB Category 2,…………

The “T/T” shows the value of -0.09 in SSB Category 2,…………

The value (e.g., 4.43 in SSB Category) is large.

Is it intercept or other?

Author Response

Comments and Suggestions for Authors

The authors conduct a cross-sectional study and aimed to evaluate the association between SSB intake/T2D-associated risk SNPs and fasting glucose in the Chilean population. The studied sample consisted of 3321 subjects (men, n= 1098 and women, n=2223) without T2D.

Comments:

1.

Table 1, SSB category

Sample sizes are inconsistent for SSB category: 686+1064+710+368+80= 2908<3321 subjects. 

Sample sizes are also inconsistent for SSB category among men (983<1098 subjects) and women (1925<2223 subjects).

Table 1, Glycemia≥100 mg/dl

Sample sizes are also inconsistent for Glycemia≥100 mg/dl:683/0.2318=3129<3321 in overall, 297/0.3=980<1098 in men, 338/0.1765=1916<2223 in women and 980+1916=2896<3129 or 3321 in combined men and women.

The sample sizes should be need to recheck in this study.

R: The inconsitency of samples sizes were the results of two issues. The submitted manuscrit contained a an old version of Table 1, by mistake, and the changes in samples sized due to quality filtering were poorly explained. There are two relevant samples sized in the study. There are 3321 non-diabetics that are described in Table 1, of which 2,895 with full SSB information and genotyping rate ≥80% were used for all association analyses. The new manuscript includes the correct version of Table 1 and an improved paragraph explaing the data filtering and sample-sizes (P3:L114-L118). In addition, we added a flowchart in the new Figure 1 to better explaing samples sizes. Sample sizes were rectified in each table.

2.

SSB consumption has been related to impaired fasting glucose (IFG) [7–9] glucose intolerance (GI), insulin resistance (IR) [10–12] and high risk of type 2 diabetes.

R: We did not find a question or commen in this quote. However, we revised and updated our references in page 2 lines 47-49:

SSB consumption has been related to impaired fasting glucose (IFG) [7,8], glucose intolerance (GI), insulin resistance (IR) [11–13], and high risk of type 2 diabetes (T2D) [14-16]

2.3 Selection of T2D risk SNPs

Single Nucleotide Polymorphisms (SNPs) for T2D risk were obtained from a Genome Wide Association Studies of (GWAS) of T2D from previous GWAS and meta-analyses 135 (SNPs were selected by statistical significance of ≤3E-8.  

R: The wording of the paragraph was corrected (see P4:L153-158).

Similarly, selected T2D-associated risk SNPs are instrumental variable (Mendelian randomization).

Maybe they have been clearly investigated the relationship of fasting glucose.

R: We did not understand this comment. However, if the reviewer is asking us to perform a Mendelian Randomization test, this is beyound the scope of the current study. Our objective was to determine if the reported associations replicate in an admixed Latin American population and if they are affected by sugary drinks intake. It is not our intent to determine causality in these associations.

3.

Authors showed the effect modification (interaction) of SSB consumption and T2D-associated risk SNPs on the fasting glucose.

Additionally, effect size or attributable risk should be reported (proportion of variance explained).

R: We have now calculate the proportion of the variances that is explained by each SNP and added these results in Supplementary Table 8. The procedure is described in “2. Materials and Methods” (P5: L219-222).

4.

Table 2

TCF7L2 rs7903146 Genotypes shown 2 “T/T”; it should be typing error.

R: thank you for this observation. It was a typing error that is corrected in the new version of Table 2.

The TCF7L2 rs7903146 “T/T”(maybe C/C) shows the value of 4.43 in SSB Category 2, 4.20 in SSB Category 3………………………….

The “C/T” shows the value of -0.006 in SSB Category 2,…………

The “T/T” shows the value of -0.09 in SSB Category 2,…………

The value (e.g., 4.43 in SSB Category) is large.

Is it intercept or other?

R= Yes, the beta values in the first genotype (no risk alleles) correspond to the intercept because they are the references genotypes. The leyend of Table 2 was improved to better describe this.

Reviewer 2 Report

This is a GWAS study including 2223 females and 1098 males’ participants. Various SSB category and characteristics were recorded and compared among females and males. Stratifications by the SSB category results were made regarding different genotypes. They found that the association between SSB intake and fasting glucose in the Chilean population without diabetes is modified by T2D genetic susceptibility.

This is an interesting study with some new findings in this area of research. The sample size of subjects is large for analysis. However, I nevertheless have the following comments that required to be addressed.

  1. The study design should be specified in this study. The authors should clarify this concern. For example, add a flow chart presented as a figure.
  2. The statistical methods used and described very well. How is the post hoc power analysis?
  3. To my knowledge, the imputation can be used to estimate genotypes under reference samples to increase accuracy. How does the authors to conduct imputation of this study?
  4. How is the variance component test in this study? The authors should clarify this concern.
  5. We know the SSB was associated with fasting glucose in population without diabetes. Do other populations have similar results? The authors should clarify this concern.
  6. How is the annotation of TCF7L2 and MTNR1B genes?
  7. Please conduct sensitive analysis through false discovery rate.
  8. Lastly, the authors only briefly discuss limitations, but didn’t discuss false-positive results. They should add some common limitations of GWAS to limitation section.

Author Response

This is a GWAS study including 2223 females and 1098 males’ participants. Various SSB category and characteristics were recorded and compared among females and males. Stratifications by the SSB category results were made regarding different genotypes. They found that the association between SSB intake and fasting glucose in the Chilean population without diabetes is modified by T2D genetic susceptibility.

This is an interesting study with some new findings in this area of research. The sample size of subjects is large for analysis. However, I nevertheless have the following comments that required to be addressed.

  • The study design should be specified in this study. The authors should clarify this concern. For example, add a flow chart presented as a figure.

R: We acknowled that a flowchart would much improve the clarity of our design and welcome this suggestion. A flowchart was added in new Figure 1 and an improved paragraph explaining data filtering and sample-sizes (P3:L112-L116).

  • The statistical methods used and described very well. How is the post hoc power analysis?

R: We now included power estimations using an on line tool https://clincalc.com/stats/power.aspx Results are shown in the new Supplementary Table 9. The results showed sufficient power, given our samples size, as long as the genotypes were frequent and genotyping rate was good. This was indicated in Results (P8: L506-508) and in Discussion (P12: L687-689).

  • To my knowledge, the imputation can be used to estimate genotypes under reference samples to increase accuracy. How does the authors to conduct imputation of this study?

R: genotype imputation can only be used when using microarrays, which generates hundreds of thousands of genotypes per individual. The present study produced genotypes for only 150 ancestry-informative SNPs and 17 risk SNPs.

  • How is the variance component test in this study? The authors should clarify this concern.

R: we don’t understand what the concern is. If the reviewer is asking for a Principal Components Analysis based on genotypes to account for genetic structure, this is not possible with the data at had (it can only be done with genome-wide data). However, our 150 ancestry-informative SNPs (AIMs) allowed us to estimate ancestry components, which were included in the models to account for genetic structure.

We calculate the proportion of the variances that is explained by each SNP and added these results in Supplementary Table 8. The procedure is described in “2. Materials and Methods” (P5: L219-222).

  • We know the SSB was associated with fasting glucose in population without diabetes. Do other populations have similar results? The authors should clarify this concern.

R: In the first paragraph of Discussion, we indicate that previous results have been inconsistent about this association, although, in average, they then to show a positive association between SSB and fasting glucose. However, we have now improved the wording of this paragraph to make this point clearer.

  • How is the annotation of TCF7L2 and MTNR1B genes?

R= The full names or both genes were added in P7: L266-275.

  • Please conduct sensitive analysis through false discovery rate.

R: We performed the sensitivity analysis and added our results P10:L320-330 and supplementary tables 10-13.

  • Lastly, the authors only briefly discuss limitations, but didn’t discuss false-positive results. They should add some common limitations of GWAS to limitation section.

R: We did not perform a GWAS, therefore we do not suffer from its limitations. Instead, we assessed association with a limited number of SNPs that were previously reported to be associated with T2D in at least one study. Therefore, we are indeed addressing a limitation of GWAS studies (false discoveries) by replicating their findings in an independent cohort. We report the limitations associated with our design in the discussion.

Round 2

Reviewer 1 Report

Authors should be provided the general characteristics of participants (n=2895, not n=3321) in Table 1, because of the study title “The association between fasting glucose and sugar sweetened beverages intake is greater in Latin Americans with a high polygenic risk score for type 2 diabetes mellitus” focus on the polygenic risk score for type 2 diabetes mellitus.

Author Response

Table 1 was updated as requested. in the process, we identified that XXX of the 2895 individuals had missing genotype information. So the actual samples size used in the study was 2828. Both Tables and Figure 1 were modified accordingly.

Reviewer 2 Report

No Further comments. Thanks for your efforts on revision.

Author Response

You you for your revision.